# Generating Data In Planning: SAS⁺ Planning Tasks of a Given Causal Structure

**Michael Katz** and **Shirin Sohrabi**

IBM T.J. Watson Research Center
1101 Kitchawan Rd, Yorktown Heights, NY 10598, USA
michael.katz1@ibm.com, ssohrab@us.ibm.com

## Abstract

The need for data in planning has long been established, by, e.g., machine learning based approaches. The existing data, however, is quite limited. There exists only a relatively small amount of hand-crafted planning domains, mostly introduced through International Planning Competitions. Further, this collection of domains is not necessarily diverse: many of these domains are some variants of the transportation problem.

In this work, we alleviate the shortage in existing planning tasks by automatically generating tasks of a particular causal structure. Given any graph $G$, we show how to create a SAS⁺ planning task with the causal graph isomorphic to $G$. We create a large collection of planning tasks by randomly generating graphs of various structural restrictions and creating planning tasks for these graphs, but also, more importantly, we provide the community with a tool that allows for on-demand generation of additional, possibly larger tasks. Our experimental evaluation ensures that the generated collection is interesting for the current state of affairs in classical cost-optimal planning, showing the performance of state-of-the-art symbolic search and explicit heuristic search based planners.

## Introduction

Since the first planning tasks encoded in STRIPS language back in 1971 (Fikes and Nilsson 1971), data, a.k.a. planning tasks, was the corner stone and one of the main drivers of research in planning. With the beginning of International Planning Competitions (IPC) in 1998 (McDermott 2000) came the increase in the availability of planning tasks, with the current estimate of slightly over 70 domains, including some variants in different formalisms. All these domains are hand-crafted, although some correspond to machine translation from a different problem (Palacios and Geffner 2009; Bonet, Palacios, and Geffner 2009; Grastien and Scala 2018; Sohrabi et al. 2018). Not only that most of these domains are hand-crafted, the collection is not necessarily diverse. Many of these domains are some variants of the transportation problem.

A major focus in classical planning was on heuristic search, with heuristics automatically obtained for planning tasks, exploiting the task structure. Few examples that explicitly exploit the causal structure include the *causal graph heuristic* (Helmert 2004) and the *structural pattern heuristics* (Katz and Domshlak 2010; Katz and Keyder 2012). Others, such as *pattern databases* (PDBs) (Edelkamp 2001) exploit the causal information in e.g., pattern selection (Haslum et al. 2007). *Merge-and-shrink heuristics* (Helmert, Haslum, and Hoffmann 2007) use the causal graph for guiding the merge process. Most existing heuristics that work on the multi-valued representation exploit the causal information in one way or another. Further, starting with the seminal work of Bäckström and Nebel (1995), the research on the complexity of planning tasks had a major focus on the characterization of planning fragments by their causal graph structure (Domshlak and Brafman 2002; Katz and Domshlak 2007; 2008; Giménez and Jonsson 2008; 2009; Katz and Keyder 2012; Bäckström and Jonsson 2013; Aghighi, Jonsson, and Ståhlberg 2015; Bäckström, Jonsson, and Ordyniak 2019), as well as some local structural characteristics, such as *k-dependence* (Katz and Domshlak 2007; Giménez and Jonsson 2012), showing these fragments to belong to a variety of complexity classes. For these two reasons, various planners performance heavily relies on the various structural characteristics of the input planning task.

The aim of this work is to generate planning tasks of a specific predefined structure. Here, we focus on the characterization of planning tasks by the structure of their causal graphs. Given a collection of multi-valued variables and a graph representing causal connections between these variables, we propose a way of generating SAS⁺ actions, initial state and a goal, in a way that the causal graph of the resulting task will match the input graph. Our aim is to be able to automatically generate a diverse collection of planning tasks, as large as needed for various purposes. One such example purpose is learning a good planner selection strategy (Sievers et al. 2019; Ma et al. 2020). Another possible purpose is an additional source of benchmarks for empirical evaluation of new planning algorithms. We test the generated collection with two modern cost-optimal planners that represent the two popular state-of-the-art approaches to cost-optimal planning. For symbolic search, we chose the planner SYMBA* (Torralba et al. 2014), winner of the sequential optimal track of International Planning Competition 2014 and one of the planners in the winning portfolio of IPC 2018

(Katz et al. 2018). For heuristic search, we chose $A^*$ with LM-cut heuristic (Helmert and Domshlak 2009), a component of many modern heuristics search based planners. Our experiments confirm that (i) the generated collection is challenging for both heuristic search and symbolic search based planners, and (ii) there is no clear dominance to any of the techniques.

The rest of the paper is structured as follows. We start with introducing the planning formalism and the notation used throughout the paper. We then move to construction, where we first describe various causal graph structures and the way these graphs can be constructed, and then describe the construction of planning tasks given a causal graph. Next, we present the experimental evaluation, including describing the way we have created our collection. Finally, we discuss the related work, and conclude with the summary of our results and future work.

## Preliminaries

A $\text{SAS}^+$ *planning task* (Bäckström and Nebel 1995) is given by a tuple $\langle \mathcal{V}, A, s_0, s_* \rangle$, where $\mathcal{V}$ is a set of *state variables*, $A$ is a finite set of *actions*. Each state variable $v \in \mathcal{V}$ has a finite domain $dom(v)$. Each pair $\langle v, \vartheta \rangle$ of variable $v \in \mathcal{V}$ and its value $\vartheta \in dom(v)$ is called a *fact*. By $F_v$ we denote the set $\{\langle v, \vartheta \rangle \mid \vartheta \in dom(v)\}$ of facts for the variable $v$, and the set of all facts is denoted by $F := \bigcup_{v \in \mathcal{V}} F_v$. A (partial) assignment to the variables $\mathcal{V}$ is called a *(partial) state*. Often it is convenient to view partial state $p$ as a set of facts with $\langle v, \vartheta \rangle \in p$ if and only if $p[v] = \vartheta$. For a partial assignment $p$, $\mathcal{V}(p) \subseteq \mathcal{V}$ denotes the subset of state variables instantiated by $p$. Partial state $p$ is *consistent* with state $s$ if $p \subseteq s$. We denote the set of states of a planning task by $S$. $s_0$ is the *initial state*, and the partial state $s_*$ is the *goal*. Each *action* $a$ is a pair $\langle pre(a), eff(a) \rangle$ of partial states called *preconditions* and *effects*. By $prv(a)$ we denote the part of the precondition that corresponds to variables that do not participate in action's effects, $prv(a) = \{\langle v, \vartheta \rangle \in pre(a) \mid v \notin \mathcal{V}(eff(a))\}$, also called *prevail condition*. An *action cost* is a mapping $C : A \to \mathbb{R}^{0+}$. An action $a$ is applicable in a state $s \in S$ if and only if $pre(a)$ is consistent with $s$. Applying $a$ changes the value of $v \in \mathcal{V}(eff(a))$ to $eff(a)[v]$. The resulting state is denoted by $s[\![a]\!]$. An action sequence $\pi = \langle a_1, \ldots, a_k \rangle$ is applicable in $s$ if there exist states $s_0, \cdots, s_k$ such that (i) $s_0 = s$, and (ii) for each $1 \leq i \leq k$, $a_i$ is applicable in $s_{i-1}$ and $s_i = s_{i-1}[\![a_i]\!]$. We denote the state $s_k$ by $s[\![\pi]\!]$. $\pi$ is a plan iff $\pi$ is applicable in $s_0$ and $s_*$ is consistent with $s_0[\![\pi]\!]$. We denote by $\mathcal{P}(\Pi)$ (or just $\mathcal{P}$ when the task is clear from the context) the set of all plans of $\Pi$. The cost of a plan $\pi$, denoted by $C(\pi)$ is the summed cost of the actions in the plan.

A central role in what follows is played by a standard structure in classical planning, called *causal graph* (Helmert 2004). The causal graph $CG_\Pi$ of a task $\Pi$ is a digraph with vertices $\mathcal{V}$. An arc $(v, v')$ is in $CG_\Pi$ iff $v \neq v'$ and there exists an action $a \in A$ such that $(v, v') \in [\mathcal{V}(eff(a)) \cup \mathcal{V}(pre(a))] \times \mathcal{V}(eff(a))$. For an action $a$, by $E_a$ we denote the set of all such arcs, and by $E_{A'}$ we denote the union of all sets of arcs $E_a$ for $a \in A'$.

Another structure typically used in planning for computing relaxation based heuristics is *relaxed planning graph* (Hoffmann and Nebel 2001), which is a layered graph of facts and actions, describing action application in the planning task, under value accumulating semantic. The layers are added until a fixpoint is reached, that is no new fact can be achieved. The first fact layer $F_1$ thus corresponds to the facts from the initial state, and the last layer is also a fact layer, and it is equal to the preceding fact layer. Each action layer $A_i$ consists of all actions from $A$ that are applicable in $F_i$, that is $A_i = \{a \in A \mid pre(a) \subseteq F_i\}$. The next fact layer $F_{i+1}$ is then constructed by adding to $F_i$ all facts achieved by the actions in $A_i$, namely $F_{i+1} = F_i \cup \bigcup_{a \in A_i} eff(a)$.

Finally, $\text{SAS}^+$ representation is often not provided directly, and is *translated* from $\text{STRIPS}$ representation (Helmert 2006). The multi-valued variables in $\text{SAS}^+$ then correspond to invariant groups of pairwise mutually exclusive facts (mutexes), where exactly one such fact is true in any state reachable from the initial state. Each such invariant group over $\text{STRIPS}$ facts corresponds to a set of facts *at most one* of which can be true in any given state that is reachable from the initial state. If there exist such states where no facts are true, then an additional value is added, representing that none of the facts in the invariant group is true.

## Construction

We start our construction by defining a graph to be served as the causal graph of the constructed task. For that, we focus here on the following causal graph structures: *chain*, *directed chain*, *fork*, *inverted fork*, *star*, *bipartite graph*, *directed bipartite graph*, *tree*, *polytree*, and *directed acyclic graph*, *complete graph*, and *random graph*. For some of these structures, namely *directed chain*, *fork*, *inverted fork*, and *complete graph* the graphs are fully defined by the number of nodes (modulo automorphisms). In other cases, we introduce randomness into the graph construction. In what follows, we first describe how we handle these cases, and then how a task with the given causal structure is constructed.

**Directed Bipartite Graph:** A full directed bipartite graph is constructed by first randomly partitioning the nodes into left and right and then introducing an edge from each node on the left to each node on the right.

**Bipartite Graph:** A full undirected bipartite graph is constructed by first randomly partitioning the nodes into left and right and then introducing an edge from each node on the left to each node on the right, and vice versa.

**Directed Chain:** A *directed* chain of $n$ nodes $v_1, \ldots, v_n$ is created by adding the edges $(v_i, v_{i+1})$ for each $1 \leq i < n$.

**Chain:** An *undirected* chain of $n$ nodes $v_1, \ldots, v_n$ is created as follows. For each $1 \leq i < n$, we randomly decide whether to add an edge $(v_i, v_{i+1})$, with probability $p$. If no such edge is added, we add the edge $(v_{i+1}, v_i)$ and if the edge $(v_i, v_{i+1})$ was added, we decide with probability $p$ whether to add the edge $(v_{i+1}, v_i)$.

**Tree:** A directed tree of $n$ nodes $v_1, \ldots, v_n$ is constructed by choosing for each node $v_i$ a parent randomly out of the

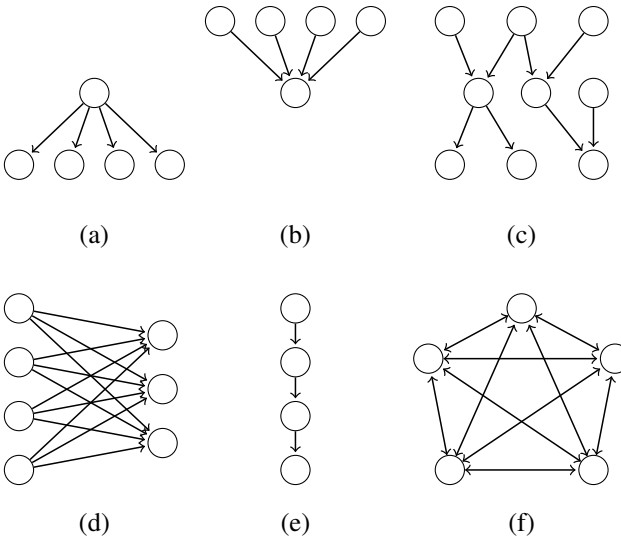

Figure 1: Selected causal graph structures: (a) fork, (b) inverted fork, (c) polytree, (d) directed bipartite graph, (e) chain, and (f) complete graph.



nodes $v_1, \ldots, v_{i-1}$.

**Polytree:** For a polytree, we start with a tree constructed as above, and then for each edge switch its direction with probability $p$.

**Directed Acyclic Graph:** A directed acyclic graph of $n$ nodes $v_1, \ldots, v_n$ is constructed by choosing for each node $v_i$ at least one parent randomly out of the nodes $v_1, \ldots, v_{i-1}$. We do that by going over all the preceding nodes and deciding with probability $p$ whether to add an edge from the preceding node to the current node. If no edges were added, we repeat until at least one edge is added for each node (except the first one).

**Random Graph:** For each pair of nodes $v_i$ and $v_j$ we randomly decide whether to add a directed edge from $v_i$ to $v_j$.

**Fork:** A fork is a directed tree with all non-root nodes being leafs, with their parent being the root node. A fork over nodes $v_1, \ldots, v_n$ is created by adding the edges $(v_1, v_i)$ for each $1 < i \leq n$.

**Inverted Fork:** An inverted fork is a directed polytree with one leaf node and all non-leaf nodes being roots, with their only child node being the leaf node. An inverted fork over nodes $v_1, \ldots, v_n$ is created by adding the edges $(v_i, v_1)$ for each $1 < i \leq n$.

**Star:** A star structure has one central node with all other nodes connected with the central node only. A star over nodes $v_1, \ldots, v_n$ is created as follows. For each $1 < i \leq n$, we randomly decide whether to add an edge $(v_1, v_i)$, with probability $p$. If no such edge is added, we add the edge $(v_i, v_1)$ and if the edge $(v_1, v_i)$ was added, we decide with probability $p$ whether to add the edge $(v_i, v_1)$.



---

**Algorithm 1** Construction of a planning task according to a given causal graph structure.

**Input:** Graph $G = (\mathcal{V}, E)$, number of facts $n \geq 2|\mathcal{V}|$

1: Partition $n$ into $|\mathcal{V}|$ values $d_v \geq 2$ such that $\sum_{v \in \mathcal{V}} d_v = n$
2: $F_v \leftarrow \{\langle v, \vartheta \rangle \mid 0 \leq \vartheta < d_v\}$ for all $v \in \mathcal{V}$
3: $s_0[v] \leftarrow 0$ for all $v \in \mathcal{V}$
4: $k \leftarrow 0$
5: $A \leftarrow \emptyset$
6: $F_k \leftarrow s_0$
7: **while** $|F_k| < n$ **or** $E \setminus E_A \neq \emptyset$ **do**
8:     $k \leftarrow k + 1$
9:     $m_k \leftarrow$ number of new facts for layer $k$
10:     $F_k, A_{k-1} \leftarrow$ CREATELAYER$(m_k, F_{k-1}, A)$
11:     $A \leftarrow A \cup A_{k-1}$
12: Select $s_* \subseteq F_k$ such that $\forall v \in \mathcal{V}, |s_* \cap F_v| \leq 1$
               and $s_* \cap (F_k \setminus F_{k-1}) \neq \emptyset$
13: **return** $\Pi = \langle \mathcal{V}, A, s_0, s_* \rangle$

14: **function** CREATELAYER$(m, F, A)$
15:     $A' \leftarrow \emptyset$
16:     $F' \leftarrow F$
17:     **while** $|F' \setminus F| < m$ **or** $(m = 0$ **and** $E \setminus E_{A \cup A'} \neq \emptyset)$ **do**
18:         $a \leftarrow$ CREATEACTION
19:         **if** $E_a \subseteq E$ **then**
20:             **if** $E_a \setminus E_{A \cup A'} = \emptyset$ **and** Random$(p)$ **then**
21:                 **Continue**
22:             $A' \leftarrow A' \cup \{a\}$
23:             $F' \leftarrow F' \cup \mathit{eff}(a)$
24:     **return** $F', A'$

---

**Complete Graph:** A complete graph over nodes $v_1, \ldots, v_n$ is created by adding the edges $(v_i, v_j)$ and $(v_j, v_i)$ for all $1 \leq i < j \leq n$.

Figure 1 exemplifies selected graph structures. Having described how a causal graph for the future planning task is constructed, we now switch to the next step, showing how to construct a planning task with that causal graph.

## Planning Task Construction

Given a graph $G = (\mathcal{V}, E)$ and a number of facts $n \geq 2|\mathcal{V}|$, we construct the SAS$^+$ planning task $\Pi = \langle \mathcal{V}, A, s_0, s_* \rangle$ with the causal graph $G$ as follows. First, we choose the domain size $d_v \geq 2$ for each of the multi-valued variables $v \in \mathcal{V}$ and assume w.l.o.g. the values to be $dom(v) = \{0, \ldots, d_v - 1\}$. The variables represent sets of mutually exclusive facts (mutexes), and each such set corresponds to one of the two types of mutexes, namely, either exactly one or at most one of the values is true in all reachable states. We randomly decide which variables belong to which category. For the variables that represent the at-most-one case, we dedicate the last domain value to represent the case when none of the other facts are true. Next, w.l.o.g. we assume $s_0[v] = 0$ for all $v \in \mathcal{V}$. Then, we construct the actions, in layers, while constructing the relaxed planning graph. Fi-

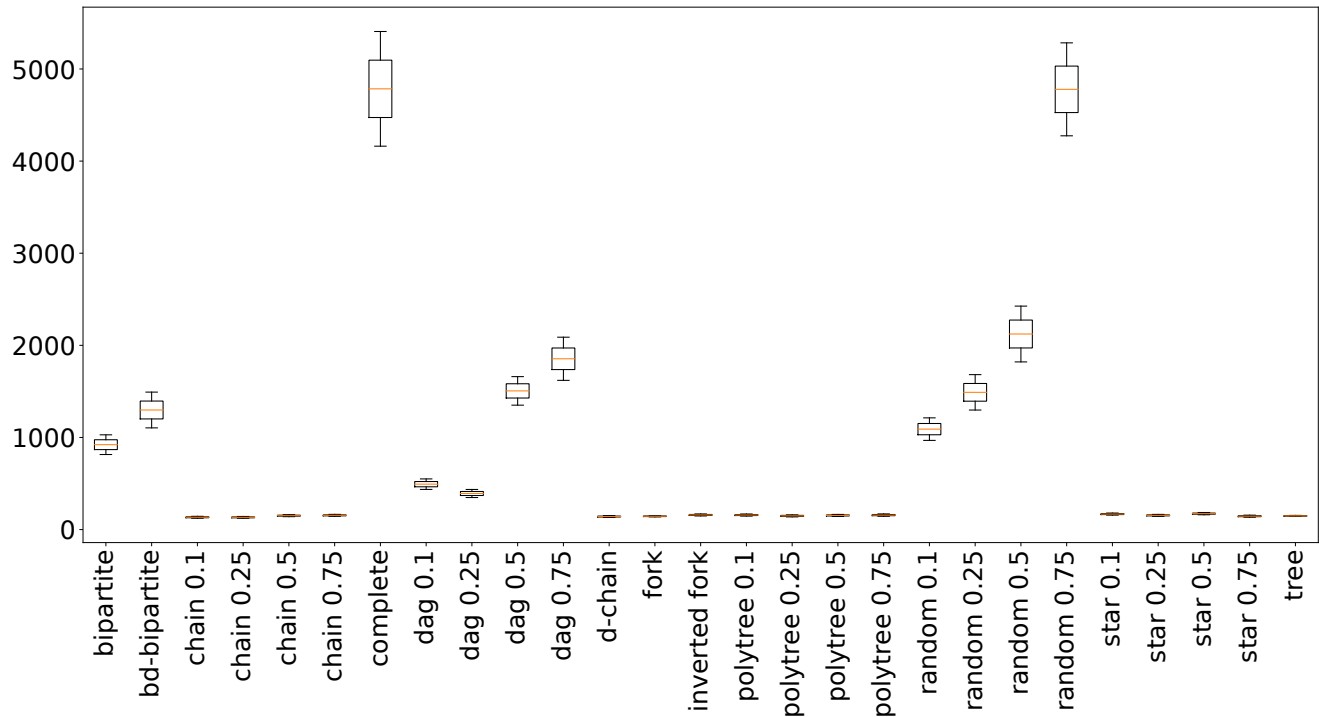

Figure 2: Mean generation time and 95% confidence intervals for each collection of tasks.

nally, the goal is chosen from the last fact layer of the relaxed planning graph, making sure that at least one of the chosen facts is unique to the last fact layer and that at most one fact is chosen per variable. In what follows, we describe how actions are constructed. Starting with the initial state as the first fact layer $F_0$, we create actions for an action layer $L_i$ by

(I) selecting a subset of facts from the fact layer $F_i$, ensuring at most one fact is selected per variable,

(II) partitioning the selected set of facts into prevail condition and non-prevail precondition, and

(III) choosing for all[1] the variables of the precondition facts a different value as its effect.

The constructed action is checked against the graph $G$, ensuring that it contributes only edges that exist in $G$. If not, the action is discarded. Additionally, if the constructed action does not add any new causal edges and does not achieve new facts, we randomly decide whether to keep it.

The generic approach to action construction described above can be adapted to enforce particular properties. We discuss three such cases in detail.

(A) The first case is enforcing the action to achieve at least one new fact. For that, one can ensure in steps (I) and

---

[1]While SAS$^+$ representation does not require to specify the precondition when the effect is specified, in order to ensure maintaining variables as mutexes of facts, we restrict ourselves here to always specifying the precondition in such cases.

(II) that the precondition includes facts for some variables $v \in \mathcal{V}$ that are not fully covered by the fact layer $F_i$ (that is $F_v \setminus F_i \neq \emptyset$), and in step (III) to choose one of these facts $F_v \setminus F_i$. Note that this can be done without adding any edges to the causal graph, if a single fact is chosen in step (I).

(B) The second case is enforcing the preconditions to include atoms from $F_i \setminus F_{i-1}$, enforced in step (I).

(C) The third case is enforcing adding a particular edge $\langle v, v' \rangle$ to the causal graph. This can be done by ensuring that $\langle v, \vartheta \rangle$ and $\langle v', \vartheta' \rangle$ are chosen in step (I), and in step (II) at most one of these facts is chosen for the prevail condition.

We randomly independently decide whether to enforce the options (A)-(C) and whether to add edges to the causal graph. Note that not all combinations are always possible. In such cases, an action is not constructed in that iteration. Each layer is constructed until a sufficient number of new facts $F_{i+1} \setminus F_i$ is added. The construction is stopped when all facts were achieved and all edges from $G$ are reflected in the causal graph of the constructed planning task. The latter is enforced in the last layer. The goal is then randomly chosen from the last layer according to step (I), ensuring that at least one of the facts is not achieved before the last layer, analogously to how a precondition of an action is chosen when enforcing the option (B). Algorithm 1 describes the construction of a planning task from the given graph $G$, where the function CREATEACTION creates a single action,

randomly choosing among the options described above.

**Theorem 1** *Given $G$ and $n$, Algorithm 1 terminates in time polynomial in $|G|$ and $n$ and returns a planning task with the causal graph $G$.*

**Proof:** The proof follows from the fact that in line 18 of Algorithm 1, for some of the options for action creation must eventually hold $E_a \subseteq E$ and $E_a \setminus E_{A \cup A'} \neq \emptyset$. Therefore, CREATELAYER terminates and returns a layer with $m$ new facts or, if $m = 0$, with $A'$ such that $E_{A \cup A'} = E$. As there are only a constant number of options, a new fact is achieved or a new causal graph edge is covered in time $O(1)$ and therefore CREATELAYER terminates in time $O(m + |E|)$. Since at least one new fact is added in each layer, Algorithm 1 terminates in time $O(n|E|)$. Since the while loop in line 7 terminates only when $E \setminus E_A = \emptyset$ and actions $a$ are added to $A$ only if $E_a \subseteq E$, when the algorithm terminates we have $E_A = E$, and therefore the causal graph of the returned task $\Pi$ is exactly $G$. $\qquad\square$

In order to create a PDDL task, the SAS$^+$ task is then translated to the STRIPS fragment of PDDL, ignoring the facts that correspond to the last value of the variables representing the at-most-one case. PDDL preconditions are taken from SAS$^+$ preconditions, add effects are taken from SAS$^+$ effects, and delete effects are taken from non-prevail preconditions. Note that if the tasks are translated from STRIPS back to SAS$^+$, there is nothing that enforces that the same mutex groups will be detected, as different planners implement different translation procedures. Thus, the causal graph structure is not necessarily preserved by translating to STRIPS and back to SAS$^+$.

## Experimental Evaluation

We start by constructing the benchmark set, as described in the previous section. Our benchmark set was generated as follows. For each of the causal graph structures mentioned above, and a value in $[0.1, 0.25, 0.5, 0.75]$ for edge probability (if needed), we create a collection of tasks. This results in 27 collections in our case, with 7 causal graph structures that do not consider edge probabilities and 5 causal graph structures that do. For each such collection, we generate 512 instances by uniformly choosing the number of atoms (4 variants), variables (4 variants), goal variables (4 variants), maximum prevail size (2 variants), maximum effect size (2 variants), and the upper bound on the minimum number of atoms per layer (2 variants). Thus, our constructed benchmark set consists of 13824 generated planning tasks. The benchmark set is available at https://github.com/IBM/structural-benchmarks-PDDL. To give a general impression of typical generation time, Figure 2 shows the mean generation time and 95% confidence intervals for each collection. It is worth mentioning that while in most collections task generation is typically quick, in some collections, such as *complete* and *random*, it can be quite time consuming. We note that these causal structures are somewhat less interesting. Nonetheless, we have decided to include these collections in our generated set.

| Collection | Comp | PDBs | Scorp | LM-cut | SYMBA$^*$ |
|---|---|---|---|---|---|
| bipartite | 105 | 73 | **182** | 180 | 59 |
| bd-bipartite | 96 | 78 | **137** | 107 | 72 |
| chain 0.1 | 358 | 321 | **395** | 372 | 282 |
| chain 0.5 | 349 | 327 | **389** | 359 | 288 |
| chain 0.25 | 409 | 345 | **444** | 428 | 309 |
| chain 0.75 | 320 | 304 | **391** | 333 | 261 |
| complete | 123 | 117 | **153** | 129 | 110 |
| dag 0.1 | 113 | 94 | **179** | 166 | 77 |
| dag 0.5 | 65 | 45 | 108 | **121** | 25 |
| dag 0.25 | 66 | 50 | **134** | 108 | 15 |
| dag 0.75 | 73 | 62 | **113** | 110 | 32 |
| d-chain | 382 | 341 | **416** | 391 | 296 |
| fork | 350 | 321 | **421** | 395 | 308 |
| inverted fork | **436** | 393 | 356 | 386 | 357 |
| polytree 0.1 | 325 | 253 | **386** | 343 | 224 |
| polytree 0.5 | 323 | 269 | **373** | 338 | 254 |
| polytree 0.25 | 367 | 268 | **405** | 379 | 245 |
| polytree 0.75 | 365 | 276 | **421** | 408 | 260 |
| random 0.1 | 54 | 38 | **167** | 95 | 32 |
| random 0.5 | 57 | 52 | **117** | 88 | 34 |
| random 0.25 | 97 | 89 | **153** | 129 | 83 |
| random 0.75 | 56 | 39 | **81** | 50 | 35 |
| star 0.1 | 226 | 130 | **272** | 224 | 125 |
| star 0.5 | 257 | 144 | **333** | 272 | 139 |
| star 0.25 | 219 | 114 | **290** | 248 | 115 |
| star 0.75 | 172 | 132 | **263** | 199 | 134 |
| tree | 364 | 277 | **433** | 402 | 245 |
| Sum (13824) | 6127 | 4952 | **7512** | 6760 | 4416 |

Table 1: Per-collection coverage of state-of-the-art planning systems: Complementary (Comp), planning-PDBs (PDBs), Scorpion (Sc), as well as $A^*$ with LM-cut heuristic and SYMBA$^*$ planner. Bolded results indicate the best coverage in a collection and overall.

Our evaluation of the constructed set aims at understanding whether the set is sufficiently challenging for modern cost-optimal planners. Therefore, we have selected the top-performing cost-optimal planners from the most recent International Planning Competition (IPC) 2018: Complementary (Franco et al. 2018), Planning-PDBs (Moraru et al. 2018), and Scorpion (Seipp 2018). We excluded the portfolio planner Delfi (Katz et al. 2018), and included instead its top performing components: the symbolic planner SYMBA$^*$ (Torralba et al. 2014) and explicit heuristic search with LM-cut heuristic (Helmert and Domshlak 2009),[2] both with $h^2$ mutex detection (Alcázar and Torralba 2015). The planners

---

[2]While the components of Delfi also use symmetry based pruning (Domshlak, Katz, and Shleyfman 2012) and partial order reduction (Wehrle and Helmert 2014), here we do not use these pruning techniques.

| Collection | Comp | PDBs | Scorp | LM-cut | SYMBA* |
|---|---|---|---|---|---|
| bd-bipartite | 66 | 66 | 66 | 66 | 61 |
| chain 0.1 | 6 | 6 | 6 | 6 | 5 |
| chain 0.5 | 8 | 8 | 8 | 8 | 7 |
| chain 0.25 | 19 | 19 | 19 | 19 | 20 |
| chain 0.75 | 30 | 30 | 30 | 30 | 21 |
| complete | 108 | 108 | 108 | 108 | 104 |
| fork | 25 | 25 | 25 | 25 | 27 |
| inverted fork | 1 | 1 | 1 | 1 | 8 |
| random 0.5 | 40 | 40 | 40 | 40 | 33 |
| random 0.25 | 50 | 50 | 50 | 50 | 46 |
| star 0.5 | 14 | 14 | 14 | 14 | 16 |
| star 0.25 | 0 | 0 | 0 | 0 | 1 |
| star 0.75 | 14 | 14 | 16 | 14 | 14 |
| Sum other | 154 | 154 | 154 | 154 | 154 |
| Sum all | 535 | 535 | 537 | 535 | 517 |

Table 2: Per-collection number of instances that proved to be unsolvable.

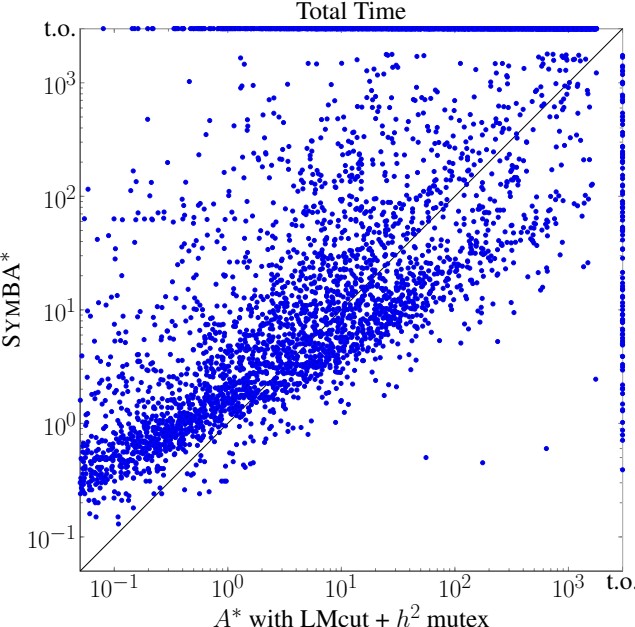

Figure 3: Total time comparison of $A^*$ with LM-cut heuristic to SYMBA* planner.

are run on the entire constructed benchmark set, with the timeout of 30 minutes and memory limit of 7.6GB allocated to each run. The experiments were performed on Intel(R) Xeon(R) CPU E7-8837 @2.67GHz machines.

Table 1 shows per-collection aggregated coverage comparison of the selected planners. Each task in a collection contributes a value of 1 to the coverage if it was either solved by the planner or the planner was able to prove the task to be unsolvable. Otherwise, the task contributes 0. Separately, Table 2 depicts the number of tasks in each collection that were proved to be unsolvable. Note that the tested planning systems perform very similarly in terms of unsolvability detection. In contrast, when looking at tasks solved, the tested planning systems perform very differently.

Going beyond aggregated coverage results, and focusing on two planners with the lowest total time on average – LM-cut and SYMBA*, Figure 3 shows the per-task total time for these planners. For the tasks solved by both approaches within the time bound, there is no clear advantage to any of the planners. Looking at the timeouts, there are 2963 cases where SYMBA* times outs but LM-cut does not, and 103 cases where LM-cut times out but SYMBA* does not.

We observe that for each of the tested planners, in each of the collections, there still remains a significant number of tasks not solved. Further, while some causal graph structures correspond to seemingly easier planning tasks, at least for the tested planners, there is a significant number of tasks in each collection that were not solved by any of the tested planners: from 54 in *chain 0.25* to 427 in *random 0.75*, with the average of 219 tasks in a collection, 5917 tasks overall. Clearly, the generated tasks are challenging for the state of the art in cost-optimal classical planning.

## Related Work

The idea of generating domain models as well as specific planning tasks has been explored in planning community, with a major focus on learning domain models from traces, for classical planning (e.g., (Yang, Wu, and Jiang 2007; Zhuo et al. 2010; Tian, Zhuo, and Kambhampati 2016)) and HTN planning (e.g., (Hogg, Muñoz-Avila, and Kuter 2008; Hogg, Kuter, and Muñoz-Avila 2010; Hogg, Muñoz-Avila, and Kuter 2016)). The work on learning domain models often assumes an existence of a complete model where the plan traces or plan examples are generated from. Some aspects of these domain models are then learned or reconstructed from successful plan traces. Some examples include learning action preconditions (Zhuo et al. 2009), or refine incomplete action descriptions (Zhuo, Nguyen, and Kambhampati 2013).

Probably a more related to our current work is the work on generating problem instances for CSP/SAT problems (e.g., (Achlioptas et al. 2000; Xu et al. 2005)). There are also several online tools/services such as the "Tough SAT Project" or "SATLIB" that generate CNF formulas encoding "difficult" problems (e.g., (Yuen and Bebel 2017; Hoos and Stützle 2000)). Producing hard satisfiable instances has several advantages one of which is to advance the research field in SAT/CSP by providing a suite of problems that can be used for evaluation of solvers. Further, these instances can be polynomial-time reduced to STRIPS in theory, but also in practice (Porco, Machado, and Bonet 2011). The authors provide a tool to translate multiple NP-complete computational problem instances (including SAT, CLIQUE, DirectedHamiltonianPath, etc.) into an NP-Complete fragment of STRIPS that they call STRIPS-1. In that fragment, the actions

are either delete-free or can be applied at most once. While the fragment is somewhat limited, the approach can be used for creating additional benchmark sets for planning. Unfortunately, the work has not yet received the attention it deserves, and the instances or the tool are not currently widely used. It is worth mentioning that our suggested approach to generating random PDDL instances is a somewhat different task than generating random CNF formula, and then translating to STRIPS. Our focus is on being able to control the causal structure of the generated problem, which is not possible with the aforementioned methods.

Another highly related work is the work on random planning tasks generation for the purpose of analyzing the phase transition in classical planning (Bylander 1996; Rintanen 2004). The authors propose a variety of models for sampling the space of STRIPS planning problem instances, exploring the possibility of phase transition at some constant ratio of the number of actions to the number of state variables. These models correspond to a constrained set of problem instances, restricting the sizes of preconditions and effects, and reducing the chances of generating trivially unsolvable tasks. Unfortunately, the proposed methods for generating tasks do not yield tasks of a desired structure and it is not clear what additional restrictions can be imposed in order to obtain such tasks.

## Summary

In this work, we have presented an approach that allows to generate planning tasks with the causal graph of a specific given structure. Further, we cast these tasks into a STRIPS fragment of PDDL, allowing using as an input to any PDDL planner that supports the STRIPS fragment, as most mature planning systems do. We have generated a benchmark set of 27 task collections characterized by the causal graph structure, with 512 tasks in each collection, summing up to 13824 ground PDDL tasks in total. Our experimental evaluation clearly shows that the generated benchmark set is challenging for both the heuristic search based and the symbolic search based planners. In the hope to facilitate further research and enable better comparison of planning tools, we make our tool publicly available to the planning community.

For future work, we intend to explore additional structural restrictions of planning tasks, such as, e.g., $k$-dependence (Katz and Domshlak 2007), as well as possibly additional causal graph structures. Further, we would like to investigate the phase transition in planning according to structural characterization of planning tasks. We conjecture that phase transition might appear at different number of actions to state variables ratios for different causal graph structures. As an additional future work, we would like to explore the usage of generating planning tasks for the purpose of learning a planner selection strategy. Finally, we would like to explore the possibility of generating lifted PDDL tasks of a meaningful causal structure. For that, we would need to understand how to characterize the ground concept of causal graph structure on a lifted level.

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
