# OpenReview forum: "Generating Data In Planning: SAS+ Planning Tasks of a Given Causal Structure"
_icaps-conference.org/ICAPS/2020/Workshop/HSDIP — HSDIP 2020_

### Official Review · AnonReviewer1 · 2020-03-31
**Interesting line of work. Evaluation could be improved, but good fit for HSDIP.**

**Rating:** 8
**Confidence:** 4

**Review:**

The paper introduces an algorithm for creating a planning task whose causal graph is isomorphic to a given input graph. It evaluates the method on several types of graphs and two prominent optimal classical planners. While the approach and the analysis could be improved and extended, the idea general approach is certainly interesting. I think the paper is a perfect fit for HSDIP.

Questions:

"[...] various planners performance heavily relies on the various characteristics of the input planning task." Which characteristics (apart from the causal graph structure) do you have in mind?

The paper cites various articles that relate problem hardness to causal graph structure. Wouldn't it be interesting to use your algorithm for confirming the findings experimentally?

I was surprised to read that you generate actions and discard them if they induce unwanted edges in the causal graph. Why can't you just generate "correct" actions?

Theorem 1: is the algorithm still polynomial even though it can be unlucky and generate unwanted actions infinitely often?

Why do you convert the SAS+ formulation to PDDL and then let the two planners convert it back? If the conversion can "destroy" the causal graph structure, why try to obtain the structure in the first place?

Traditionally, AI planning has focused on solvable instances. Do you have an idea on how to guarantee solvability of the generated instances?


Comments:

If I didn't overlook something, the paper leaves open a number of questions: e.g., how do you choose the domain sizes? How do you choose the number of new facts per layer? I think you need to present the algorithm with greater detail, including the variants for all decisions, and possibly make space for this by removing some variants and focusing on the method's core.

The paper uses only two planners to evaluate the new benchmarks. If you are preparing to submit the paper to a conference, I think that the evaluation has to greatly expanded. E.g., you will need to include more than two planners. I recommend using planners from IPC 2018 or planners that build on IPC 2018 planners. This should allow you to make an unbiased selection of state-of-the-art planners (e.g., LM-Cut has been ousted as state-of-the-art by abstraction heuristics a while ago).

It seems overkill to generate 512 instances per class. 30 instances should be enough to show difference between planners.


Minor comments:

"R^{0+}" -> R^+_0

"value accumulating semantic++s++"

[0.1, 0.25, 0.5, 0.75] -> Why not uniform?

"unsolvability is recognize++d++"

"unsolvability is recognized [...] by SymBA* in all 507 cases" -> Do you really know that none of the unsolved tasks is unsolvable?

The text states in twice (in two sentences) that there are tasks unsolved by any of the two planners. You can remove one sentence.

---

### Official Review · AnonReviewer2 · 2020-04-01
**Initial Assessment**

**Rating:** 7
**Confidence:** 4

**Review:**

This paper addresses describes an algorithm to generate a planning task with a causal graph that matches a given graph.

The paper is well written and easy to follow and tackles a problem interesting to the planning community. I think it fits HSDIP even though it is not directly related to search or heuristic questions.

Causal graph structures:
The text lists 12 of them, but only explains 7. I would expect that all 12 are explained, even if it is of the form "directed bipartite graph is a special case of the pipartite graph". Also, please make the order of presentation consistent at all 3 places: in the first list in the text, in the detailed list of definitions later in the text, and in Figure 1 (which could/should include more examples).

Translation from SAS+ to PDDL:
Given that any PDDL task can be grounded in many ways (depending on the type of reachability analysis typically performed), I think the inverse translation is non-trivial. At least to me, it is, and the paper does not explain how to derive predicates and action schemata from ground STRIPS representations which only operate on binary variables with no further semantics. What also worries me is that, as the paper states, the translation from PDDL back to SAS+ is far from being unique, and thus resulting planning tasks could have a very different causal structure than the one originally intended.

Algorithm 1:
While fairly easy to follow, I'd suggest to walk the reader through the algorithm while explaining the construction in the text. In the current form, matching pseudocode to text was not always easy.

Construction:
- it was first unclear to me why it is important that SAS+ facts have a meaning, until reading at the end that the task it translated back to PDDL
- prevail conditions vs preconditions: not defined
- why is it guaranteed that an action compatbile with (I)-(III) exists/can be defined?
- how is m chosen? Why is m=0 possible? Why, if m=0, it is guaranteed that all missing edges of E can still be covered through adding new actions (probably the same question as the previous one)?

Evaluation:
- picking two planners seems to be not enough for judging the quality of results: I would suggest to run at least a few IPC 2018 planners (due to being containerized, this should be possible fairly easily)
- what worries me is the generation of unsolvable tasks. I see that there is probably no easy way of preventing this, but this really lowers the quality of the resulting benchmark set. Please report the number of unsolvable tasks (as far as it is possible to know, using other techniques suitable for detecting unsolvability) and the part of it detected by the evaluated planners. Also report the number of tasks not solved by the planners.

---

### Comment · AnonReviewer2 · 2020-08-04
**No further comments**

While I titled my first review "initial assessment", I realize that it already includes everything I had to comment on after reading the paper. Since there was no reaction or a revised submission, I also don't have anything further to add.

---

> ### Author Response · Authors · 2020-08-04
> **The current version is a revised submission**
>
> We apologize for not replying to the comments, but we did submit a revision where we tried our best to accommodate for reviewers' comments.

---

> > ### Comment · AnonReviewer2 · 2020-08-04
> > **In this case, I have more comments**
> >
> > In this case, I have a few more comments after reading the revised submission.
> >
> > Introduction:
> >
> > - merge-and-shrink: I would consider citing the journal paper (JACM 2014)
> > rather than the original conference paper
> >
> > - complexity of planning tasks: I have the feeling there could be further
> > (newer) related work by Bäckström and/or Jonsson that are also based on the
> > causal graph
> >
> > Construction:
> >
> > - there is one "and" too many in the list of causal graph structures
> >
> > - bipartite graphs: rather than copying the text, how about saying that this is
> > like directed bipartite graphs but with the inverse edges in addition?
> >
> > - some of the graph structures mention probability p, others don't, even though
> > they also depend on randomization. I would assume that p always applies for all
> > random choices, doesn't it?
> >
> > - are forks/inverted forks special cases of trees/polytrees with depth 1? Could
> > be an easier definition.
> >
> > - Figure 1 could be moved on the next page
> >
> > - "prevail" was not defined
> >
> > - it is a bit confusing that in addition to the construction rules (I) --
> > (III), the "options" (A) -- (C) can be added onto that. Some of the options
> > sound like enforcing what is required according to rules (I) -- (III).
> >
> > - "in such cases, an action is no constructed in that iteration": I thought
> > that this was forbidden. How is the polynomial runtime still guaranteed?
> >
> > - similarly, if the number of new facts for layer k, m_k, is chosen so that it
> > is "sufficient", what does it mean? m_k = 0 seems to be allowed, but why does
> > the runtime guarantee then still hold? is m_k = 0 allowed for other layers than
> > the last one when there are still edges from E missing in the constructed task?
> > also, m_k shouldn't be too large, I guess?
> >
> > - besides this number m_k, there are is at least one other choice (besides the
> > purely random choices regarding graph structures) that are not really
> > described: how are values partitioned into variables? maybe there are further
> > design decisions that should be mentioned?
> >
> > - the notation "E_A" was not defined
> >
> > - "PDDL preconditions are taken from SAS+": this was not so obvious to me. I
> > understand that the technique generates *ground* PDDL, but still, more details
> > should be given here.
> >
> > - footnote 2: put it *after* the comma, i.e. ",^2 " rather than "^2, "
> >
> > - I wondered how long it takes to generate the planning tasks. Especially since
> > the algorithm relies on creating the "right" actions in order to finish
> > computing a layer, I wonder if this cannot take long time when being unlucky.

---

> > > ### Author Response · Authors · 2020-08-08
> > > **Thank you for additional comments**
> > >
> > > Additional complexity results: thank you, indeed we have found additional work that will be cited:
> > > https://www.ida.liu.se/~chrba09/Papers/jair13-v47.pdf
> > > https://dl.acm.org/doi/10.5555/2888116.2888165
> > > https://www.ida.liu.se/~chrba09/Papers/socs2018.pdf
> > >
> > > Polynomial runtime: once the precondition variables are chosen and partitioned into prevail and non-prevail parts, the edges that the action contributes to the causal graph are uniquely defined. For a constant size precondition upper bound, there is a polynomial number of such choices and therefore an action that agrees with the causal graph will be selected after a polynomial number of iterations. We can elaborate on that in the paper.
> > >
> > > Time it takes in practice: our implementation was not aimed at speedy execution and can be significantly sped up. In fact, we have made a few recent optimizations that were not reflected in the reported benchmark set.  Indeed, when unlucky, on very large instances, for random and complete causal graphs, a task can take a long time to be constructed. For the experiments reported in the paper, the maximal time was slightly over 6 hours. Our latest implementation is significantly faster and can be further sped up. Nevertheless, 30% of the tasks were constructed in under 1 minute and 85% of the tasks in under 10 minutes.

---

> > > > ### Comment · AnonReviewer2 · 2020-08-10
> > > > **Suggestion to mention the runtime to create the benchmarks**
> > > >
> > > > Thanks for the clarifications. I think it would be interesting to mention the time it takes to construct the benchmark set so that potential users of your tool know what to except if they want to generate lots of benchmarks.

---

> > > > > ### Author Response · Authors · 2020-08-14
> > > > > **Certainly**
> > > > >
> > > > > We can include an information about construction time in the final version.

---

### Comment · Program_Chairs · 2020-09-14
**Final Decision: Accept**

Dear Authors,

Thank you very much for your submission. We are happy to inform you that we have decided to accept it and we look forward to your talk in the workshop. You will receive additional information per mail in the coming days.

Best,
The HSDIP'20 team

---

### Decision · Program_Chairs · 2020-09-30

Accept